# HFDream: Improving 3D Generation via Human-Assisted Multi-view Text-to-Image Models

## Abstract

Large-scale text-to-image models have demonstrated the potential for performing text-to-3D synthesis. However, existing approaches, e.g., DreamFusion, suffer from unstable 3D optimization due to the limitations of current text-to-image models that they struggle to synthesize images from certain viewpoints even when specified in the text prompt. Obtaining a view-aligned image-text pair dataset is challenging due to the limited availability of such data, and the inherent subjectivity and ambiguity of view-alignment. In this paper, we propose to enhance text-to-3D generation by learning from human feedback for generating desired views. We generate multi-view images with the text-to-image model and engage human labelers to select a valid viewpoint. Using the human-labeled dataset, we train a reward model designed to verify whether the generated image aligns with the viewpoint specified in the text prompt. Finally, we fine-tune the text-to-image model to maximize the reward score. We find that our text-to-image diffusion models fine-tuned with human feedback, coined HFDream, consistently generate diverse viewpoints without the need for multi-view datasets created from 3D assets. This leads to high-quality text-to-3D generations with consistent geometry, when combined with view-dependent prompting in DreamFusion.

## 1 Introduction

Large-scale pre-trained text-to-image diffusion models (Saharia et al., 2022; Rombach et al., 2022; Nichol et al., 2021) have demonstrated the potential in text-to-3D generation (Poole et al., 2022; Wang et al., 2023a; Lin et al., 2023), which allows to synthesize 3D content with only textual descriptions. Notably, even without using any 3D data, providing view-dependent prompts (e.g., front view, side view, back view) (Poole et al., 2022) to the text-to-image diffusion models can guide the optimization of a 3D representation, such as Neural Radiance Field (NeRF; Mildenhall et al. 2021).

However, since current text-to-image models often fail to generate images that are aligned with the designated viewpoint, the generated 3D contents can have inconsistent 3D geometry, such as multi-face Janus problem (Armandpour et al., 2023). To address this problem, several recent works (Shi et al., 2023; Liu et al., 2023b) proposed to utilize 3D data prior, which finetune diffusion models with 3D data. While this approach enhances the geometric consistency of text-to-3D generation, the finetuned diffusion models lose sample diversity and fidelity due to the distribution shift between pretrained image data and 3D data. To overcome this fundamental limitation, we ask the following question: *How to improve the geometric consistency of 3D generation without using such 3D data?*

To properly answer this question, we utilize *learning from human feedback* framework (Lee et al., 2023; Fan et al., 2023) to create multi-view 2D datasets by filtering outputs of text-to-image models with human feedback. This provides more realistic and diverse multi-view images and eliminates the need for using a 3D asset-based dataset. We then fine-tune the text-to-image model using human-labeled multi-view datasets to generate desired views for enhancing text-to-3D generation. Specifically, our approach involves the following steps (see Figure 1). (a) We generate multi-view images using text-to-image models and collect human feedback to select a valid viewpoint. (b) With a human-labeled dataset, we train a reward model to assess whether the generated image aligns with the viewpoint specified in the text prompt. (c) We fine-tune the text-to-image model to maximize reward scores using a reward-weighted loss (Lee et al., 2023). (d) We integrate the fine-tuned text-to-image

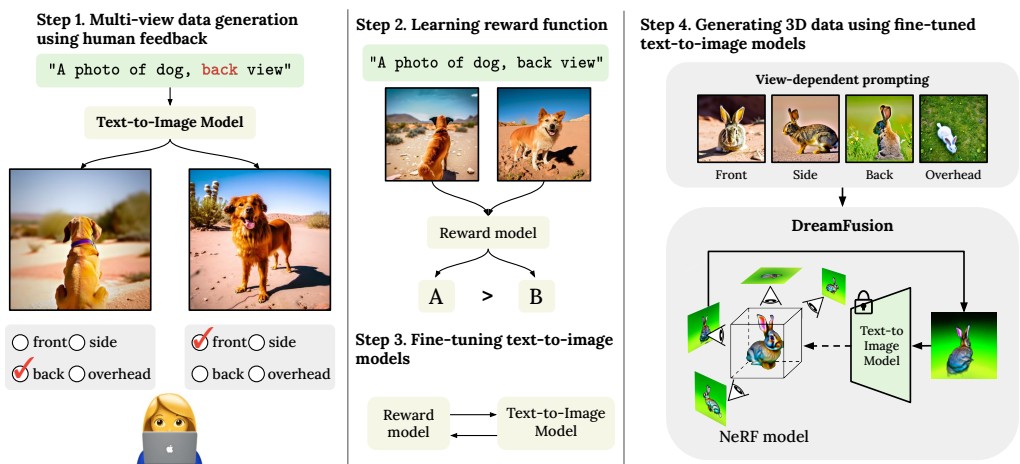

Figure 1: **Overview of our method.** HFDream consists of four stages: (1) We generate multi-view images using the text-to-image model and engage human annotators to select a valid viewpoint. (2) We train a reward model that verifies whether the image aligns with the viewpoint specified in the text prompt. (3) We fine-tune the text-to-image diffusion models by maximizing the reward through reward-weighted loss (Lee et al., 2023; Fan et al., 2023). (4) We perform text-to-3D generation by using DreamFusion (Poole et al., 2022) with our view-aligned diffusion models.

models with view-dependent prompting in DreamFusion (Poole et al., 2022) for 3D generation. Our fine-tuned model consistently generates diverse viewpoints and enables high-quality 3D generation.

We summarize our main contributions as follows:

- We propose a simple fine-tuning method to improve the multi-view generation ability of the text-to-image model using human feedback, leading to better text-to-3D generation when it is combined with DreamFusion.

- We demonstrate that our reward function predicts human assessments of view alignment more accurately than ImageReward, which is pre-trained on extensive human preference datasets.

- Fine-tuned text-to-image model, HFDream, significantly improves the learned reward scores, over the original model, implying its ability to produce view-aligned images.

- The improved ability to generate view-aligned images resulted in an improvement in text-to-3D quality. In human evaluation, HFDream achieves a 31% gain in text alignment and 26% gain in 3D quality, compared to DreamFusion.

## 2 RELATED WORK

### 2.1 TEXT-TO-IMAGE GENERATION

Diffusion models have demonstrated their capacity to effectively learn data distributions and subsequently generate high-fidelity image samples from Gaussian noise by mastering the denoising process (Sohl-Dickstein et al., 2015; Ho et al., 2020; Song et al., 2020). Furthermore, significant advancements have been made in the development of various guidance methods (Kawar et al., 2022; Ho & Salimans, 2022; Bansal et al., 2023; Kim et al., 2022), which enhance conditional image generation. The integration of diffusion models with pre-trained text encoders (Radford et al., 2021; Raffel et al., 2020) has led to impressive results in text-to-image generation (Ramesh et al., 2022; Rombach et al., 2022; Saharia et al., 2022).

### 2.2 VIEW-CONSISTENT 2D GENERATION FOR TEXT-TO-3D GENERATION

Recent work has demonstrated the capability of text-to-image models to generate multi-view images and optimize implicit neural networks such as NeRF (Mildenhall et al., 2021) and InstantNGP (Mildenhall et al., 2021; Müller et al., 2022) as representations for 3D scenes. DreamFusion (Poole et al.,

2022), for instance, proposed a new method of training NeRF using a score distillation loss obtained from a text-to-image diffusion model. Subsequent studies Wang et al. (2023a); Lin et al. (2023) have explored various distillation techniques.

In the context of these 2D-lifted text-to-3D generation methods, ensuring view consistency in generated images is crucial. However, achieving this consistency remains challenging due to inherent viewpoint biases in current text-to-image models. Addressing this challenge, PerpNeg (Armandpour et al., 2023) introduced a perpendicular gradient sampling algorithm, effectively eliminating undesired attributes (e.g., wrong viewpoints) indicated by negative textual prompts. However, PerpNeg is limited in that it does not explicitly address the view-consistency problem in text-to-image generation. Another line of research centers on novel view synthesis based on the image-to-image diffusion model. These approaches can be seamlessly integrated with text-to-image generation to produce view-consistent images. Liu et al. (2023a) and Shi et al. (2023), in particular, leverage the Objaverse dataset (Deitke et al., 2023) to fine-tune image-to-image models. These models are conditioned on a single source view image and a novel viewing direction to generate new view image. Additionally, Liu et al. (2023b) introduced an image-to-image diffusion model capable of generating multiple views within a single reverse process. Despite achieving view consistency, these methods require the acquisition of costly 3D assets. Moreover, the process of fine-tuning generative model on object-centric 3D assets may potentially compromise the generalization of the pre-trained diffusion models.

## 2.3 ALIGNING TEXT-TO-IMAGE MODELS WITH HUMAN FEEDBACK

Reinforcement learning from human feedback (RLHF) has demonstrated the alignment capabilities inherent in large-scale language models (Stiennon et al., 2020; Ouyang et al., 2022; Bai et al., 2022). Recent work has also applied such methods to improve the alignment of text-to-image models (Lee et al., 2023; Fan et al., 2023; Xu et al., 2023; Kirstain et al., 2023). Lee et al. (2023) showed that simple fine-tuning pre-trained models using the reward-weighted likelihood can enhance the alignment of text and generated image pairs concerning attributes such as object colors, counts, and backgrounds specified within the text. Xu et al. (2023) and Kirstain et al. (2023) have scaled up this direction by collecting large-scale human feedback datasets and training reward functions from human preference data. Building on these advancements, Fan et al. (2023) and Black et al. (2023) have introduced reinforcement learning (RL) algorithms for fine-tuning text-to-image models using reward functions. Motivated by the successes of these methods, we propose a fine-tuning method with human feedback for improving the multi-view generation, leading to better text-to-3D generation.

## 3 MAIN METHOD

In this section, we outline our approach to fine-tuning a text-to-image diffusion model for generating desired views and enhancing text-to-3D generation. First, we generate multi-view images using the text-to-image model and engage human labelers to select a valid viewpoint (Section 3.1). Subsequently, we train a reward model specifically designed to verify whether the generated image aligns with the viewpoint specified in the text prompt (Section 3.2). Following this, we update the text-to-image models to maximize the reward score through a reward-weighted loss (Section 3.3). Finally, we integrate the fine-tuned text-to-image models with view-dependent prompting in DreamFusion (Poole et al., 2022) for 3D generations (Section 3.4).

## 3.1 MULTI-VIEW DATA COLLECTION WITH HUMAN FEEDBACK

**Multi-view image dataset.** To create a multi-view image dataset, we use text prompts generated by combining one viewpoint from {front, side, back, overhead} with an object. Specifically, given viewpoint A and object B, we form them into natural language using the template: `A photo of a , <A> view`; for example, `A photo of a dog, front view`. In total, we use 162 text prompts for 18 objects and 9 scenes, and about 1200 annotated images were gathered per direction. Because the diffusion model has a low chance of successful generations, especially for backside views, we generate more images until we gather enough ground truth images. Details regarding labeling can be found in Appendix B.

**Human feedback collection.** We ask human annotators to select a valid viewpoint for the generated image from the options {front, side, back, overhead}. In contrast to previous methods that rely solely

on binary (good/bad) feedback (Lee et al., 2023) or preferences (Xu et al., 2023; Kirstain et al., 2023) to assess model outputs, this allows us to gather a more fine-grained assessment of the viewpoint of generated images. Once a valid viewpoint is determined by the human annotator, we reassign the text prompt accordingly using the same template.

## 3.2 MULTI-VIEW REWARD LEARNING

Using images with a valid viewpoint, we train a reward model $r_\phi(\mathbf{x}, \mathbf{y})$ specifically tailored to assess whether the generated image $\mathbf{x}$ aligns with the viewpoint described in the corresponding text prompt $\mathbf{y}$. For the reward learning process, we augment our image-text dataset by introducing negative examples. For each image-text pair $(\mathbf{x}, \mathbf{y})$ associated with a valid viewpoint, we choose a negative image $\mathbf{x}'$, with a different viewpoint as indicated in the original text prompt $\mathbf{y}$. This process generates a dataset $\mathcal{D}^{\mathtt{human}} = \{(\mathbf{x}, \mathbf{x}', \mathbf{y})\}$. We follow the basic framework for learning rewards from preferences (Christiano et al., 2017), which is commonly applied in both the language domain (Stiennon et al., 2020; Ouyang et al., 2022) and text-to-image domain (Xu et al., 2023; Kirstain et al., 2023). The reward model $r_\phi(\mathbf{x}, \mathbf{y})$ is trained by minimizing the following negative log-likelihood loss:

$$\mathcal{L}^{\mathtt{reward}}(\phi) = -\mathbb{E}_{(\mathbf{x}, \mathbf{x}', \mathbf{y}) \sim \mathcal{D}^{\mathtt{human}}}[\log(\sigma\left(r_\phi(\mathbf{x}, \mathbf{y}) - r_\phi(\mathbf{x}', \mathbf{y})\right))],$$

where $\sigma$ denotes the logistic function. We remark that augmenting our dataset with negative images can prevent model bias and trivial guessing by balancing the training data. Text-to-image models frequently struggle with producing certain viewpoints, such as the back view, leading to dataset imbalance. This imbalance can make the reward model susceptible to trivial guessing. Our augmentation strategy involves oversampling (Chawla et al., 2002) minority samples to alleviate this issue.

In our experiments, instead of learning a reward model from scratch, we fine-tune ImageReward (Xu et al., 2023), which is an open-source reward model pre-trained on an extensive dataset of human preferences. To mitigate overfitting to our human dataset, we also incorporate the ImageReward-2k dataset from pretraining into our reward learning process.

## 3.3 FINE-TUNING TEXT-TO-IMAGE MODELS

We fine-tune a text-to-image diffusion model $p_\theta$ using learned reward model $r_\phi$ by minimizing the reward-weighted negative log-likelihood (Lee et al., 2023) with KL regularization (Fan et al., 2023):

$$\mathcal{L}^{\mathtt{t2i}}(\theta) = \mathbb{E}_{p(\mathbf{y})}\mathbb{E}_{p_{\mathtt{pre}}(\mathbf{x}|\mathbf{y})}[-r_\phi(\mathbf{x}, \mathbf{y})\log p_\theta(\mathbf{x}|\mathbf{y}) - \gamma\log p_\theta(\mathbf{x}|\mathbf{y})], \tag{1}$$

where $p(\mathbf{y})$ is the distribution of prompts, $\gamma > 0$ is the KL penalty, and $p_{\mathtt{pre}}$ denotes the initial text-to-image diffusion model. However, we empirically observe that directly using the learned reward model often makes the training unstable since the reward values vary across different viewpoints[1]. To address this issue, we introduce a simple normalization scheme which utilizes the softmax function of rewards for different viewpoints. Formally, our normalized reward is given by

$$s_\phi(\mathbf{x}, \mathbf{y}) = \frac{\exp\left(r_\phi(\mathbf{x}, \mathbf{y}) / \tau\right)}{\sum_{\mathbf{y}' \in \mathcal{Y}(\mathbf{y})} \exp\left(r_\phi(\mathbf{x}, \mathbf{y}') / \tau\right)}, \tag{2}$$

where $\tau > 0$ is a temperature parameter that controls the disparity between views, and $\mathcal{Y}(\mathbf{y})$ represents the set of text prompts with four possible viewpoints {front, side, back, overhead} for the same object in the original prompt $\mathbf{y}$. This normalization step helps stabilize the fine-tuning process by addressing the issues of varying reward values associated with different viewpoints. We replace the reward $r_\phi(\mathbf{x}, \mathbf{y})$ with normalized score $s_\phi(\mathbf{x}, \mathbf{y})$ in equation 1.

Due to the complexity of computing the exact log-likelihood for text-to-image diffusion models like Imagen (Saharia et al., 2022), Stable Diffusion (Rombach et al., 2022) and DALL-E (Ramesh et al., 2022), we employ an alternative approach. Specifically, we minimize the supervised training objective with KL-O, which is introduced in Fan et al. (2023) and corresponds to an upper bound of equation 1.

---

[1]For example, we find that text prompts with a front view usually have higher reward scores than text prompts with a back view.

Table 1: **View-aligned image generation.** We measure and compare the softmax reward score for Text-to-image generation.

| | Front view | | Side view | | Back view | | Overhead view | |
|---|---|---|---|---|---|---|---|---|
| | Seen | Unseen | Seen | Unseen | Seen | Unseen | Seen | Unseen |
| DeepFloyd-IF | 0.498 | 0.421 | 0.504 | 0.452 | 0.129 | 0.166 | 0.866 | 0.695 |
| HFDream | **0.812** | **0.590** | **0.900** | **0.646** | **0.360** | **0.339** | **0.903** | **0.862** |

### 3.4 TEXT-TO-3D GENERATION

We employ DreamFusion (Poole et al., 2022) as our main framework for generating 3D models using our fine-tuned diffusion model, HFDream. DreamFusion updates the parameters of a NeRF model (Mildenhall et al., 2021) by introducing noise to a NeRF rendering output image, which is sampled from a randomly selected camera position. Text-to-image models refine this noisy NeRF rendering through image-to-image generation, conditioned on input text prompt. The refined image is then used to train the NeRF model, thereby gradually guiding its learning process. To improve geometric consistency in the refinement step, DreamFusion uses view-dependent prompts based on the random camera position, adding a viewpoint suffix that corresponds to the camera position. For example, if the azimuth value falls within the range of [-45, 45], then a directional suffix `front view` is appended to the prompt, constructing a view-dependent prompt `A photo of <A>, front view`. While this view-dependent prompting can improve the geometric consistency of the generated 3D rendering when text-to-image models consistently generate desired views, prior text-to-image models often fail to generate images that are aligned with the designated viewpoint. This results in 3D inconsistency problems such as the Janus problem (Armandpour et al., 2023). However, HFDream mitigates these problems by generating view-aligned images.

## 4 EXPERIMENTS

### 4.1 EXPERIMENTAL SETUP

The reward model is fine-tuned from ImageReward-v1.0 (Xu et al., 2023), which was trained using over 140K expert-annotated preference pairs. Our human-labeled dataset consists of over 200K augmented preference pairs after minority oversampling. We sample 200 pairs per prompt, resulting in roughly 32K pairs. The reward model is then trained using both our dataset and ImageRewardDB 2K dataset, which has 35.5K preference pairs. Each minibatch is balanced to ensure equal sampling from both datasets. The reward model is trained with a batch size of 128 and learning rate $1 \times 10^{-5}$, and we choose the best validation checkpoint based on view-direction classification accuracy after training for 30 epochs on the validation set.

We use DeepFloyd-IF (Saharia et al., 2022) as our baseline text-to-image model, which is a pixel-level diffusion model trained on large image-text datasets. For fine-tuning, we utilize Low-Rank Adaptation (LoRA; Hu et al. 2021), which introduces trainable weights to the UNet of the diffusion model and updates only these added weights. We use a total batch size 128 and learning rate $1 \times 10^{-6}$ during fine-tuning for 5K steps. 25K images are generated per direction, which are then filtered based on the top 20% normalized rewards. For text-to-3D generation, we follow DreamFusion (Poole et al., 2022), with the backbone text-to-image model replaced by our fine-tuned text-to-image model, HFDream.

For evaluation we construct a comprehensive text prompt set to examine HFDream's ability to generate view-aligned images and enhance the geometric consistency in 3D generation. We split the evaluation set into two subgroups *seen* and *unseen*, based on whether the object in the prompt was encountered during the training of the text-to-image model. The evaluation prompt set consists of 61 distinct text prompts (23 seen text prompts and 38 unseen text prompts).

Details about the evaluation prompts can be found in Appendix A.

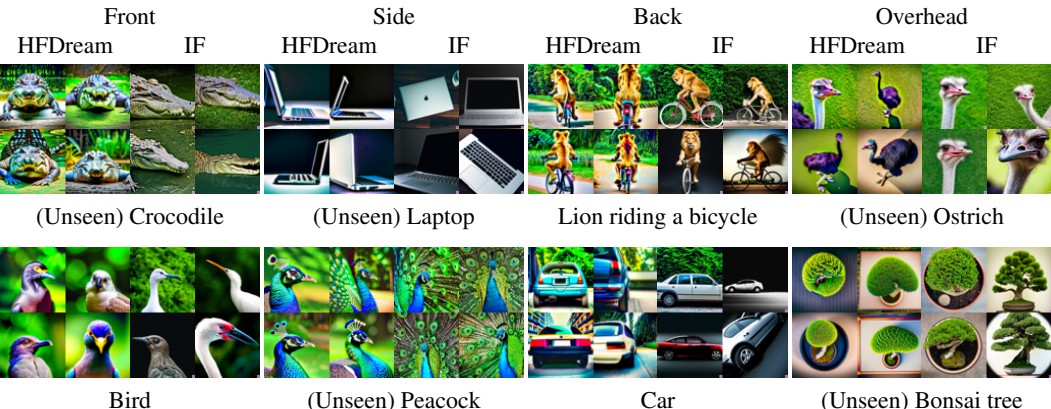

| Front | | Side | | Back | | Overhead | |
| HFDream | IF | HFDream | IF | HFDream | IF | HFDream | IF |

(Unseen) Crocodile     (Unseen) Laptop     Lion riding a bicycle     (Unseen) Ostrich

Bird     (Unseen) Peacock     Car     (Unseen) Bonsai tree

Figure 2: **Qualitative comparison** of HFDream and vanilla DeepFloyd-IF, with 4 random seeds.

## 4.2 VIEW-DEPENDENT TEXT-TO-IMAGE GENERATION

In this section, we evaluate HFDream's ability to generate view-aligned images, comparing it to the baseline model, DeepFloyd-IF. For evaluation, we create view-dependent prompts by adding viewpoint prefixes to 61 text prompts (described in Section 4.1), resulting in the generation of a total of 1952 images (8 images for each view-dependent prompt). We then report the average normalized reward score (equation 2) per direction in Table 1. While DeepFloyd-IF faces challenges in generating images from specific viewpoints such as side and back, HFDream significantly improves view-dependent generation across all directions. Notably, HFDream consistently outperforms the original model even on unseen prompts. However, its performance drop on unseen prompts (from 0.900 to 0.646 for side view) does suggest the potential need for larger and more diverse human datasets. Figure 2 indeed shows HFDream consistently achieves view-aligned text-to-image generation, regardless of whether the concept was seen during training or not, overcoming the predominant bias in the original model. For example, HFDream can generate a back view of a `lion riding a bicycle` based on novel text prompts with counterfactual concepts.

## 4.3 TEXT-TO-3D GENERATION

To evaluate the benefits of consistent multi-view generation in text-to-3D generation, we combine HFDream with DreamFusion (Poole et al., 2022) (denoted as DF-HFDream). For comparison, we also consider DreamFusion in combination with two other methods: pre-trained DeepFloyd-IF (denoted as DF-IF) and pre-trained DeepFloyd-IF with PerpNeg (Armandpour et al., 2023) (denoted as DF-PerpNeg). PerpNeg is a negative prompting algorithm that can remove unwanted attributes (undesired views in our case). For each text prompt in 61 evaluation prompts mentioned in Section 4.1, we show two 3D outputs generated based on the text prompt (one from DreamFusion with HFDream and one from the baseline) to human raters. Then human raters indicate which one is better, or tie (i.e., two outputs are similar) in terms of the 3D quality (both geometric consistency and texture detail) and text alignment, respectively. Each of the 244 3D asset (4 seeds per prompt) is evaluated by 4 independent human raters. Details about the human evaluation regarding 3D quality and Alignment with text is elaborated in Appendix C.

Figure 5a shows win-tie-lose rates across the raters. Our method demonstrates significant improvements in both 3D quality and text alignment, achieving up to 31% improvement (45%-14% win-lose rate) in text alignment and 26% improvement (51%-25% win-lose rate) in 3D quality compared to DF-IF. As shown in Figure 3, our DF-HFDream consistently produces 3D renderings with geometric consistency, whereas DF-IF often fails to generate high-quality outputs. We remark that a similar trend is observed when comparing DF-HFDream with DF-PerpNeg. DF-HFDream also mitigates the Janus problem effectively. In Figure 4, we compare the output of DF-HFDream against DF-IF, using same text prompt `A zoomed out DSLR photo of a snail`, where images from the four pre-defined viewpoints (front, back, side, and overhead) are placed in order. Notably, the baseline output exhibits the Janus problem, most prominently visible in the rightmost image (overhead view), where the snail appears to have two heads.

DF-HFDream                                    DF-IF

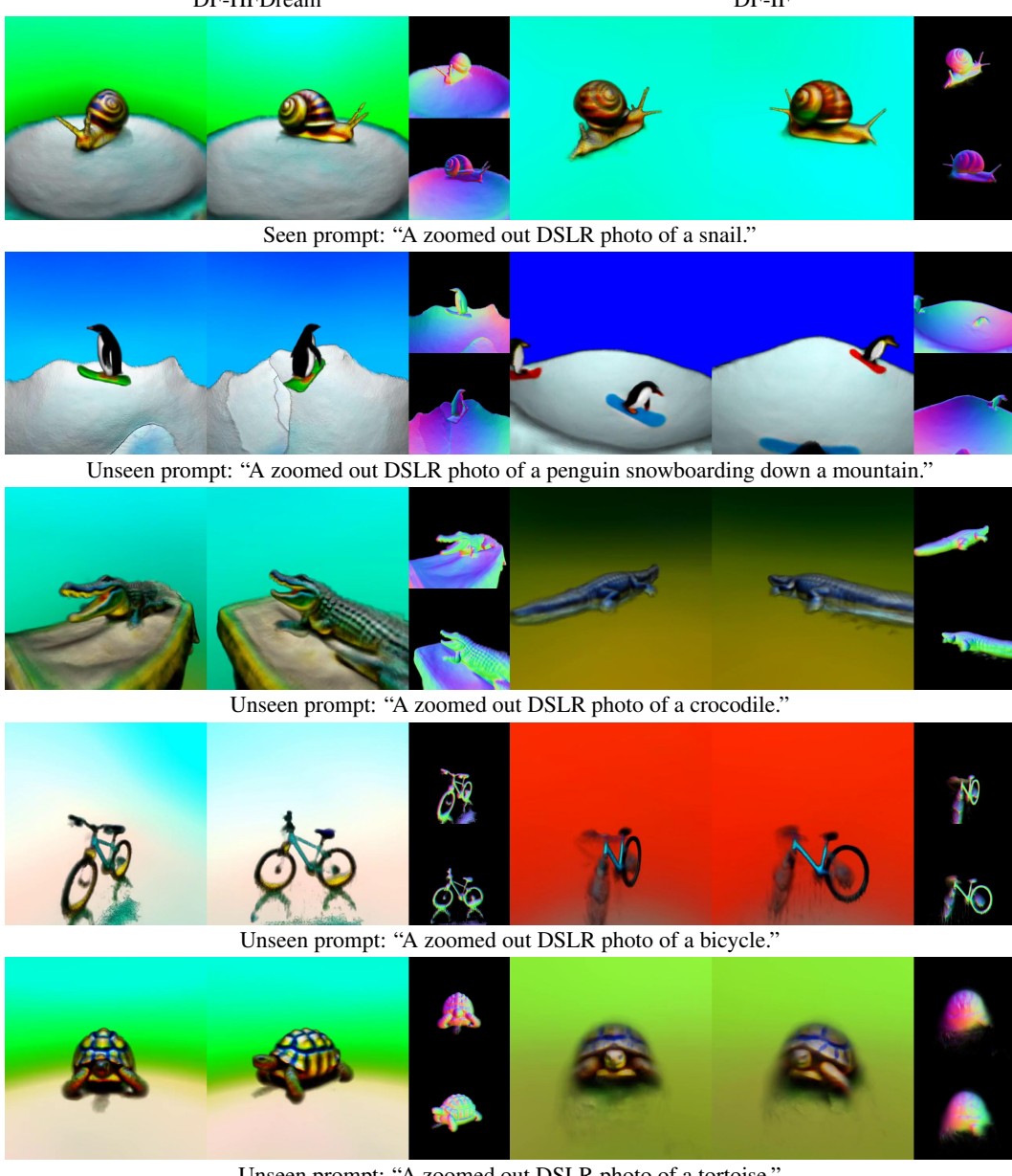

Seen prompt: "A zoomed out DSLR photo of a snail."

Unseen prompt: "A zoomed out DSLR photo of a penguin snowboarding down a mountain."

Unseen prompt: "A zoomed out DSLR photo of a crocodile."

Unseen prompt: "A zoomed out DSLR photo of a bicycle."

Unseen prompt: "A zoomed out DSLR photo of a tortoise."

Figure 3: **Qualitative comparison** of DF-HFDream and DF-IF.

**View-alignment of the output.** We also verify whether the 3D outputs are view-aligned by conducting a human survey. To this end, we select 4 rendered views of 3D output from { front, left side, right side, back} based on viewing coordinate.[2] Human raters evaluate the set of four images and judge whether all four images are view-aligned. They are instructed to skip the query when it is hard to make a clear decision and provide binary feedback: "good" if all four images are view-aligned or "bad" otherwise. We generate a total 244 3D assets from 61 prompts (4 seeds from each prompt) and each query is evaluated by 4 independent human raters. We report the average fraction of good and bad queries across the raters. As shown in Figure 5b, 50% of the outputs generated by DF-HFDream are indicated as perfectly view-aligned, while only 32% of the outputs generated by DF-IF exhibit the same level of view alignment. Although DF-PerpNeg shows better view alignment compared to DF-IF, DF-HFDream still outperforms it.

---

[2]To precisely evaluate geometric consistency, we use both left and right sides.

DF-HFDream                                                    DF-IF

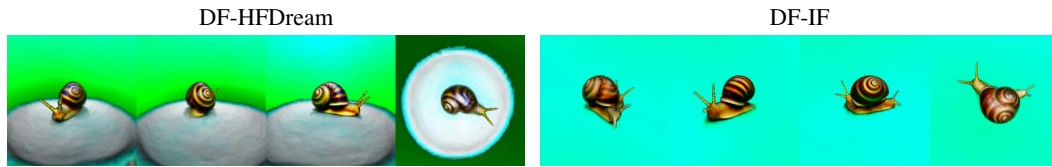

Seen prompt: "A zoomed out DSLR photo of a snail."

Figure 4: **Qualitative visualization of the Janus problem.** HFDream mitigates the Janus problem by generating view-aligned images.

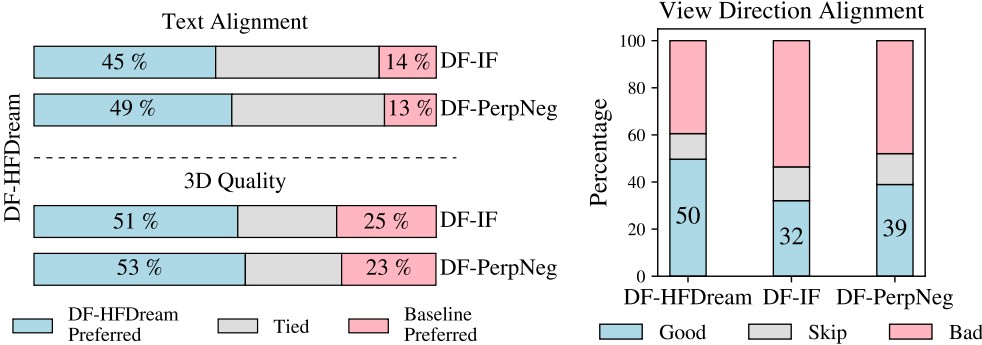

(a) Win rate of DF-HFDream against baselines.          (b) Percentage of perfectly view-aligned models.

Figure 5: **Human survey evaluation of DF-HFDream and baseline methods.** For text alignment and 3D quality, DF-HFDream consistently shows a higher win rate over baseline methods. The percentage of perfectly view-aligned generations is also significantly improved.

**Quantitative result.** We also perform quantitative evaluations using two key metrics: normalized reward scores and CLIP R-Precision (Poole et al., 2022). For normalized reward scores, we select 4 images of 3D output from each viewpoint and measure the learned reward score, which serves as a metric for 3D view alignment. For CLIP R-Precision, we measure the accuracy of the CLIP model in retrieving the input prompt from among other text prompts, utilizing image embeddings derived from RGB renderings (referred to as Color) and textureless renderings (referred to as Normal), in combination with text embeddings. Specifically, CLIP R-Precision (Color) serves as a metric for overall fidelity in 3D rendering, while CLIP R-Precision (Normal) serves as a metric for geometric fidelity. We refer to Appendix C for details of evaluation metrics. We report all evaluation metrics on both seen and unseen text prompts in Table 2, showing that DF-HFDream outperforms all baselines across, consistent with the human evaluation.

## 4.4 REWARD MODEL EVALUATION

This section evaluates the reward model's ability to predict the alignment between an image and a view-dependent prompt. We compare our fine-tuned version of ImageReward against the default ImageReward-v1.0. In the training step, reward model is guided to return a higher score for the image that is aligned better with the text prompt, and this ability is measured by using "preference accuracy". Preference accuracy counts the cases where $r_\phi(\mathbf{x}_1, \mathbf{y}) > r_\phi(\mathbf{x}_2, \mathbf{y})$ if the human prefers $\mathbf{x}_1$ over $\mathbf{x}_2$, for the same prompt $\mathbf{y}$. In addition, we measure the classification accuracy: predicting a view direction based on reward scores. Higher classification accuracy requires the reward model to return lower scores for all incorrect directions. In practice, we find the classification accuracy to be a better measure of the model's overall performance. Here, we report the validation accuracy of the reward model on its training dataset with 5K images.

In Figure 6, the original ImageReward model is unable to categorize some views at all, achieving less than 10% accuracy. In contrast, our fine-tuned version achieves around 90% accuracy for all directions, regardless of whether the object was encountered or not during the reward model training. Note that the "seen" and "unseen" sets are different from the ones we've used in 2D and 3D evaluation; in this figure, they denote whether they were seen in the training of the reward model. Finally, the preference accuracy for the default ImageReward model is still deceivingly high, which reveals that classification accuracy is a better measure of the model's performance.

Table 2: **Text-to-3D generation comparison.**

| | CLIP R-Prec. Color (↑) | | CLIP R-Prec. Normal (↑) | | Normalized Reward Score (↑) | |
|---|---|---|---|---|---|---|
| | Seen | Unseen | Seen | Unseen | Seen | Unseen |
| DF-IF | 65.3 | 64.1 | 29.8 | 33.1 | 0.480 | 0.358 |
| DF-PerpNeg | 61.3 | 56.3 | 28.8 | 31.4 | 0.481 | 0.362 |
| DF-HFDream | **70.5** | **70.2** | **40.5** | **38.8** | **0.531** | **0.413** |

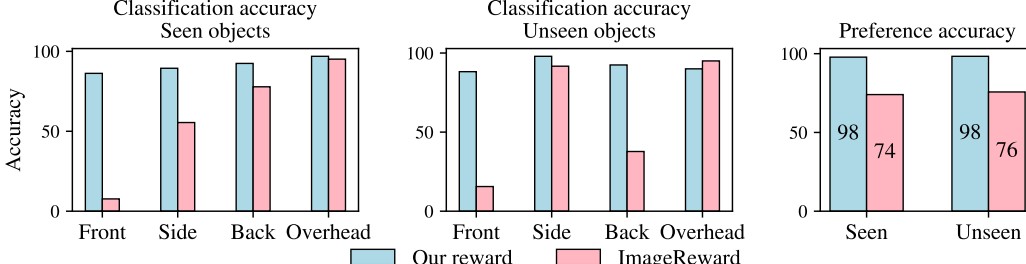

Figure 6: Comparison of our fine-tuned reward model and the ImageReward model.

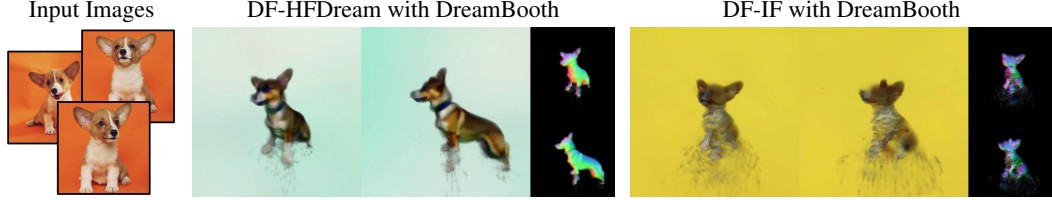

"A zoomed out DSLR photo of a sks dog."

Figure 7: **Text-to-3D DreamBooth generation.** HFDream is able to perform personalized text-to-3D generation even without specialized techniques.

### 4.5 HFDREAMBOOTH3D

An interesting application of HFDream involves fine-tuning it on personalized object datasets, using tools like DreamBooth (Ruiz et al., 2023). However, directly applying text-to-3D generation with a DreamBooth-tuned text-to-image model is often unstable due to the text-to-image model's susceptibility to overfitting (Raj et al., 2023). Consequently, such models struggle with view-dependent generation. We demonstrate that HFDream consistently generates view-dependent images, making it a promising choice for personalized text-to-3D generation after simple DreamBooth tuning, without the need for additional algorithms. We name this simple method HFDreamBooth3D, and compare it to the original DeepFloyd-IF model used in the same manner. As shown in Figure 7, HFDream successfully produces personalized 3D outputs through this straightforward and cost-efficient approach, outperforming the original DeepFloyd-IF model, which fails to deliver satisfactory results under the same conditions.

## 5 CONCLUSION

In this work, we have demonstrated that fine-tuning with human feedback can effectively improve multi-view image generation and enhance text-to-3D generation. We create multi-view 2D datasets by filtering outputs of text-to-image models with human feedback, resulting in more realistic and diverse images. Without using a 3D asset-based dataset, we efficiently fine-tune the text-to-image diffusion model for generating desired views. Our text-to-image diffusion models fine-tuned with human feedback, coined HFDream, consistently generate diverse viewpoints without the need for multi-view datasets created from 3D assets. This leads to high-quality text-to-3D generations with consistent geometry, when combined with view-dependent prompting in DreamFusion.

## ETHICS STATEMENT

Text-to-image or Text-to-3D models hold promise for various applications such as art, graphics, VR, movies, and gaming. However, current models have several limitations in generating multi-view images or rendering 3D data with consistent geometry. Our research offers a solution by enabling fine-tuning of these models to improve specific properties and align their behaviors with human intentions. However, at the same time, this process can be socially harmful depending on the human data collection and learned reward functions. For example, one can expect that users could train models to produce fake images/3D data or copy contents without permission.

To prevent this, details of human labeling and data generation should be clearly documented. It is also critical to clarify potential bias and failure cases of reward models. In this work, we provide all details of data generation and labeling instructions in Section 3, Section 4, Appendix B and Appendix C. We also discuss the limitations of our learned reward model (e.g., slightly low accuracy in unseen prompts and specific viewpoints like back) and failure cases of our fine-tuned models in Section 4. We believe that our research can provide helpful insights in aligning text-to-image and text-to-3D models with human intentions and improving their ability in image and 3D synthesis.

## REPRODUCIBILITY STATEMENT

We provide the experimental setup in detail, including how we collected human feedback data, how we trained the reward model and the text-to-image diffusion model, and the methods that we employed to evaluate our method and the baseline methods in Section 4 and Appendix A.

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

## A   EVALUATION PROMPT

In this section, we present the set of textual prompts utilized in the evaluation of HFDream's view-dependent text-to-image and text-to-3D generation capabilities. A total of 61 prompts were carefully composed for this evaluation, encompassing 23 prompts related to objects encountered during the fine-tuning process and 38 prompts featuring objects that were not part of the fine-tuning dataset. To examine HFDream's generation capability in handling textual prompts spanning a spectrum of complexity, ranging from succinct descriptions to more intricate ones, we employed two distinct templates, namely, "`A zoomed-out DSLR photo of a <object>.`" and "`A zoomed-out DSLR photo of a <object> <attribute>.`", to construct the prompts. Entire concepts used for constructing prompts are described in 3.

Table 3: **Evaluation prompts.**

|  | Simple concept | Complex concept |
|---|---|---|
| Seen prompt | bear.
bird.
cat.
dog.
fish.
lion.
pig.
rabbit.
wolf.
snail.
tiger.
car.
shoe. | bear reading a book.
lion riding a bicycle.
rabbit juggling oranges.
pig rolling in the mud.
snail crawling on a leaf.
bear watching a smartphone.
rabbit nibbling on a carrot.
bird made out of fruit.
fish made out of stone.
pig made out of salad.
-
-
- |
| Unseen prompt | giraffe.
kangaroo.
koala.
dolphin.
octopus.
penguin.
rhinoceros.
crocodile.
gorilla .
ostrich.
peacock.
seahorse.
chimpanzee.
lamp.
laptop.
camera.
toaster.
books.
bicycle.
candelabra.
bonsai tree.
easel with a painting.
tortoise.
whale.
castle.
teapot.
hot air balloon.
spider. | ostrich made out of salad.
dolphin made out of stone.
penguin made out of fruit.
polar bear playing a saxophone.
octopus riding a unicycle.
ostrich skateboarding down a hill.
dolphin playing chess with robot dolphin.
gorilla playing a piano.
penguin snowboarding down a mountain.
kangaroo wearing a boxing glove.
-
-
-
-
-
-
-
-
-
-
-
-
-
-
-
-
-
- |

## B   LABELING INSTRUCTION

Human annotators were given a single image at a time, randomly sampled from the four direction categories. Annotators then answer what direction the image is viewed from, among the four directions. Annotators were instructed to skip images with more than 1 image, images that are not aligned with the text prompt, images where the prompted object is not recognizable, or images that feel uncertain.

## C   EVALUATION DETAILS

### C.1   HUMAN SURVEY

For the evaluation of (1) 3D quality and (2) Alignment with text based on human survey, users were presented with pairs of 3D items, one generated by a baseline method and the other by HFDream.

For (1) 3D quality, users were asked to select the item that has better consistency and more detailed texture. The question posed was: "Which item has better geometric consistency and texture detail?". For (2) Alignment with text, participants chose the 3D item that is better aligned with the input text prompt. The question for this task was: "Which 3D item is well aligned with the input text prompt?".

## C.2 CLIP R-PRECISION.

CLIP R-Precision is an automated metric that measures the fidelity of the image in relation to its text input. It is the accuracy of CLIP model's retrieval of the ground truth caption among 'distractors' (captions not corresponding to the image). Poole et al. (2022) measure this value on both RGB renderings and textureless renderings, and found that using textureless renderings better measure the geometric quality, as the CLIP model is forced to retrieve the text without using any color information. Although Poole et al. (2022) used 153 captions from the COCO validation set following Dream Fields (Jain et al., 2022), we use 61 prompts introduced in 4.1. As the prompts in our evaluation set are more similar to each other, it is more challenging to retrieve the ground truth text among them. For each method including HFDream, we generate 244 3D assets (61 prompts × 4 seeds per prompt). For each 3D asset, we render the object from 120 viewpoints, each with the same elevation and equally distributed azimuth values. Then we calculate CLIP R-Precision values for every rendered views, and average them to produce averaged CLIP R-Precision for each 3D asset. We repeat this process with normal map renderings, as they contain more information about surface geometry than textureless renderings while eliminating the color information.

## D EVALUATION RESULTS OF MAGIC3D-HFDREAM.

To demonstrate that our method is compatible with any text-to-3D method, we apply our method on Magic3D (Lin et al., 2023). Magic3D consists of two steps: (1) First, a NeRF model is trained in low-resolution image space to capture coarse outlines and (2) then, it is further fine-tuned in high-resolution image space to create fine details of the model. Specifically, an open source implementation (Guo et al., 2023) of Magic3D uses DeepFloyd-IF-I-XL with a resolution of 64 for step 1, and Stable Diffusion 2.1 with a resolution of 512 for step 2. We denote this as Magic3D-IF-SD. We compare this baseline to HFDream combined with Magic3D, denoted as Magic3D-HFDream. For Magic3D-HFDream, we use HFDream-tuned DeepFloyd-IF and Stable Diffusion 2.1 for each of the two stages. As shown in Table 4, our method consistently improves both CLIP R-Precision and normalized reward scores for both seen and unseen evaluation sets. Additionally, Figure 8 shows that the Janus problem is greatly alleviated in Magic3D-HFDream. For example, the bear has 3 legs and the tortoise has 2 heads in the case of the original Magic3D method (highlighted with a yellow circle), while our method does not exhibit these issues.

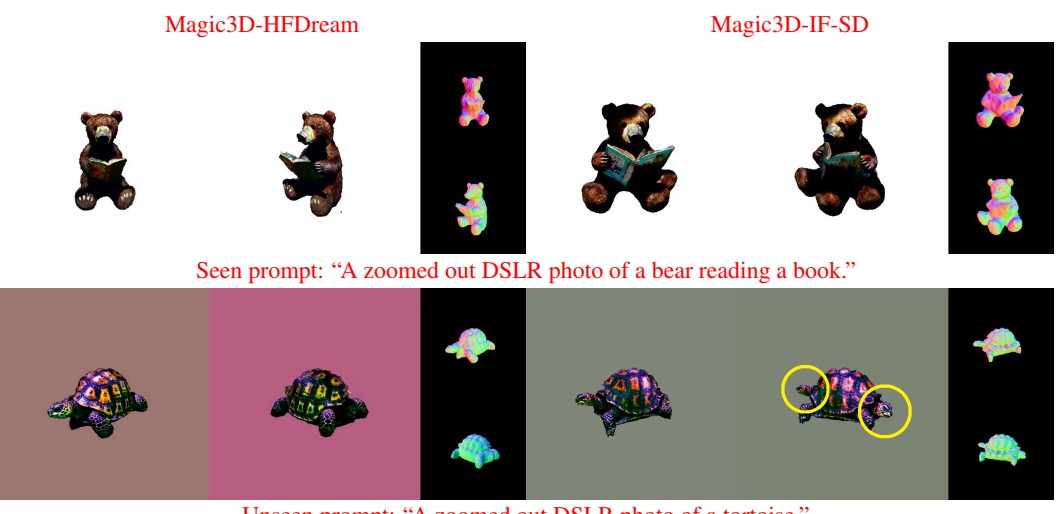

Magic3D-HFDream                                    Magic3D-IF-SD

Seen prompt: "A zoomed out DSLR photo of a bear reading a book."

Unseen prompt: "A zoomed out DSLR photo of a tortoise."

**Figure 8: Qualitative comparison** of DF-HFDream with DF-IF (above), and Magic3D-HFDream with Magic3D-IF-SD (below).

Table 4: Text-to-3D generation comparison.

| | CLIP R-Prec. Color (↑) | | CLIP R-Prec. Normal (↑) | | Normalized Reward Score (↑) | |
|---|---|---|---|---|---|---|
| | Seen | Unseen | Seen | Unseen | Seen | Unseen |
| Magic3D-IF-SD | 63.6 | 62.0 | 32.1 | 34.1 | 0.343 | 0.539 |
| Magic3D-HFDream | **65.6** | **72.9** | **43.8** | **35.9** | **0.427** | **0.578** |

## E QUALITATIVE RESULTS

### E.1 2D GENERATION RESULTS OF HFDREAM.

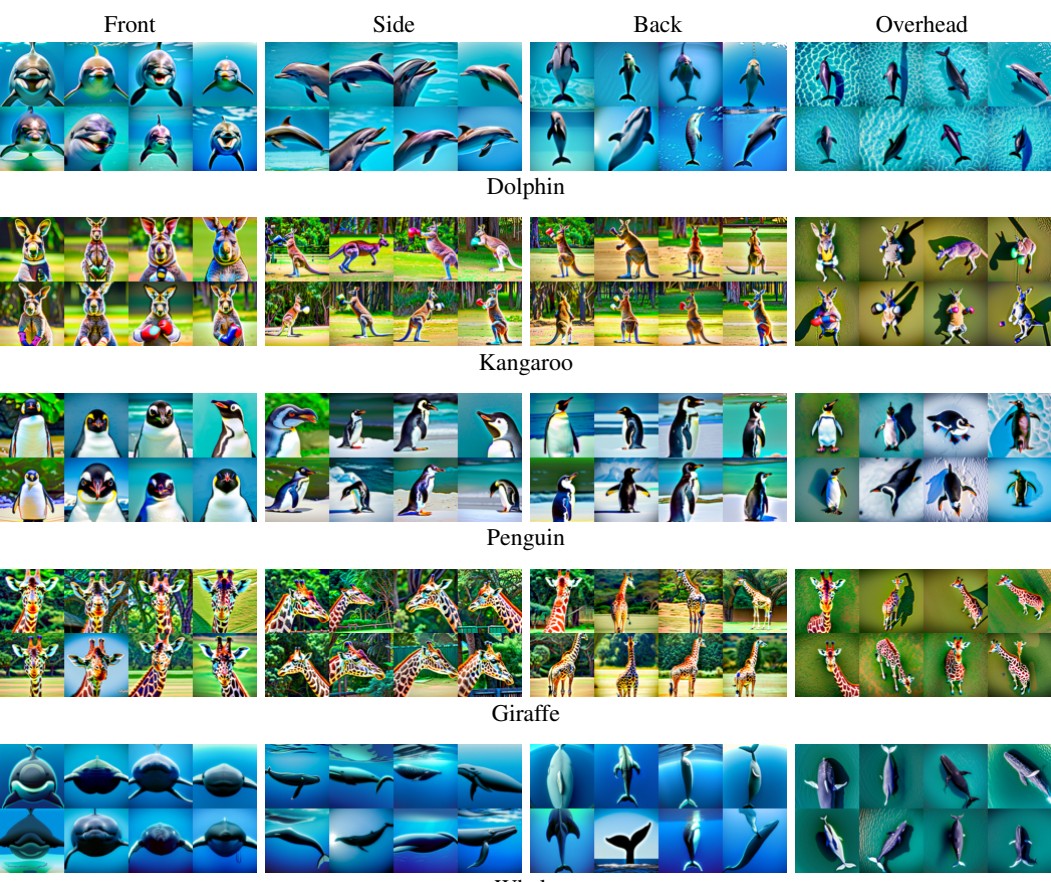

Figure 9: HFDream text-to-image generation samples for unseen objects, with 8 random seeds.

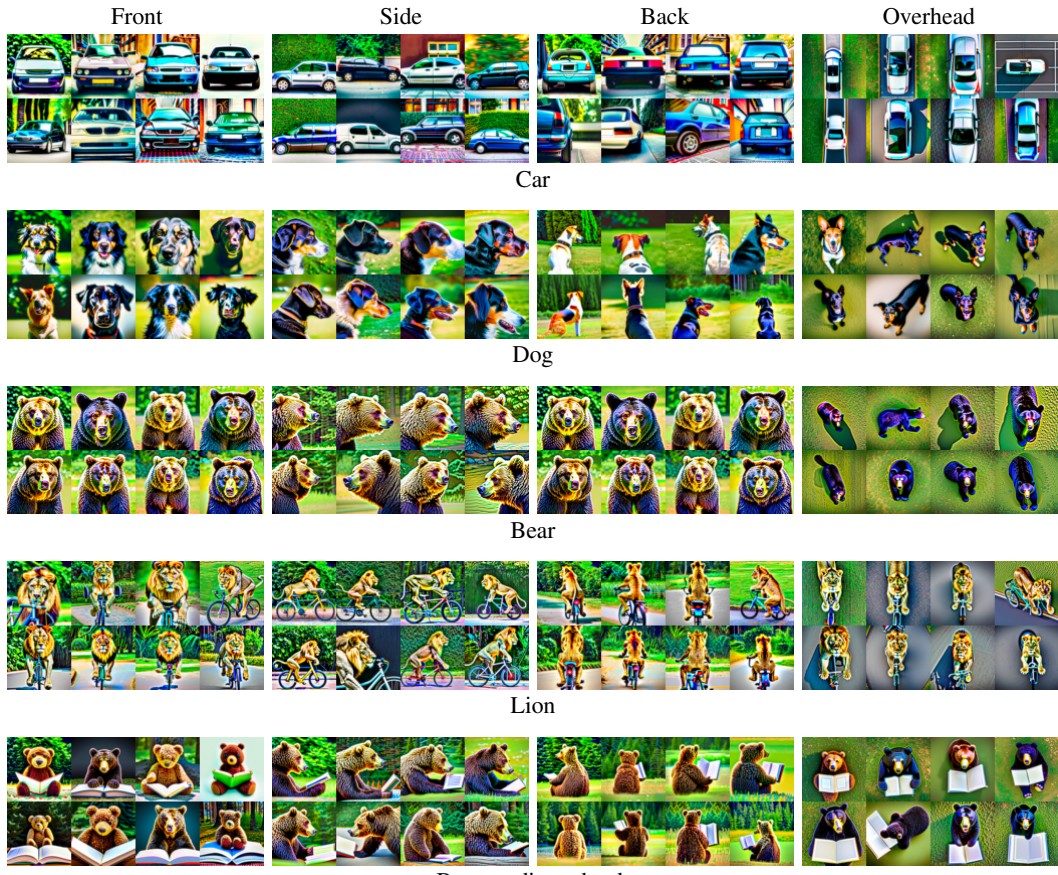

Figure 10: HFDream text-to-image generation samples for seen objects, with 8 random seeds.

## E.2 3D GENERATION RESULTS WITH DREAMFUSION-HFDREAM.

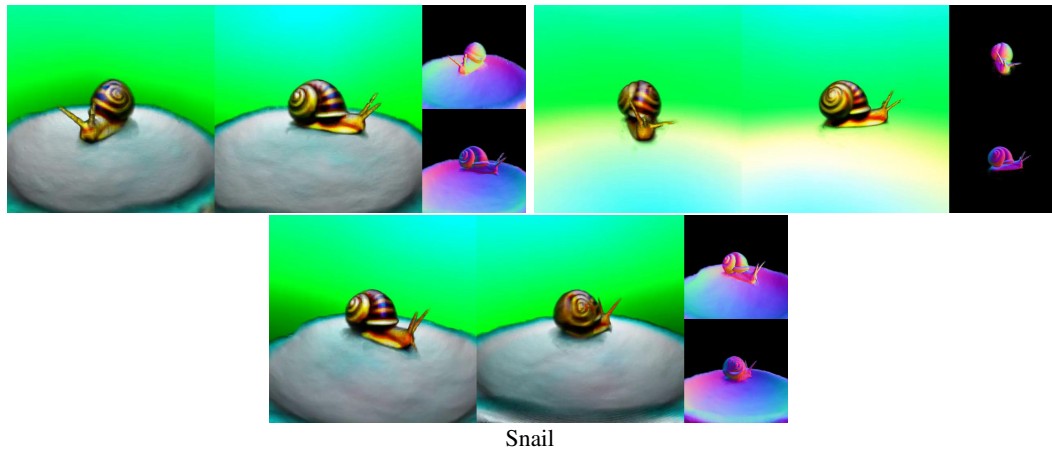

Snail

Figure 11: DF-HFDream text-to-3D generation example with 3 random seeds.

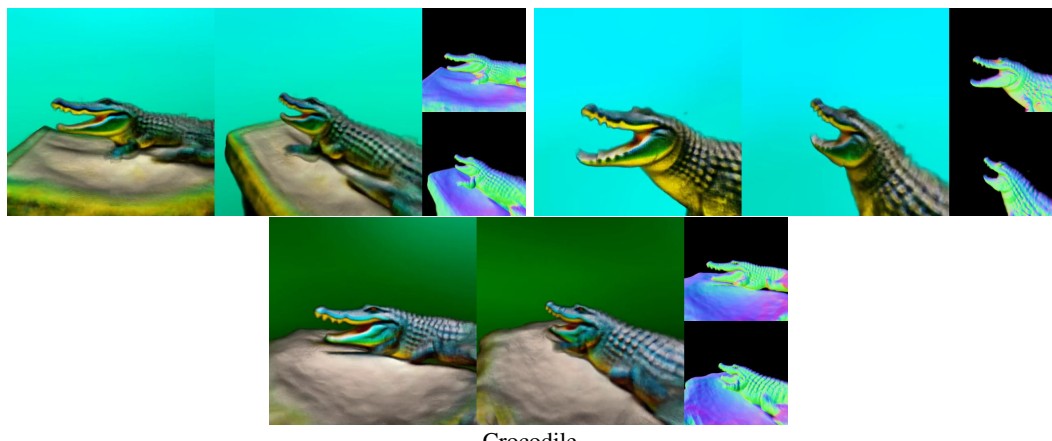

Crocodile

Figure 12: DF-HFDream text-to-3D generation example with 3 random seeds.

### E.3   QUALITATIVE COMPARISON OF HFDREAM AND VARIOUS BASELINES

We compare HFDream to various baselines including MVDream (Shi et al., 2023). The baseline images and prompts are from the MVDream [link] in Figure 13, Figure 14 and Figure 15. Overall, our Magic3D-HFDream demonstrates consistently better results in terms of geometric consistency and text alignment. For example, in Figure 13, MVDream sacrifices text alignment for geometric consistency (rocket is missing), and other methods, such as DreamFusion-IF, Magic3D-IF-SD, TextMesh (Tsalicoglou et al., 2023), ProlificDreamer (Wang et al., 2023b), have the Janus problem.

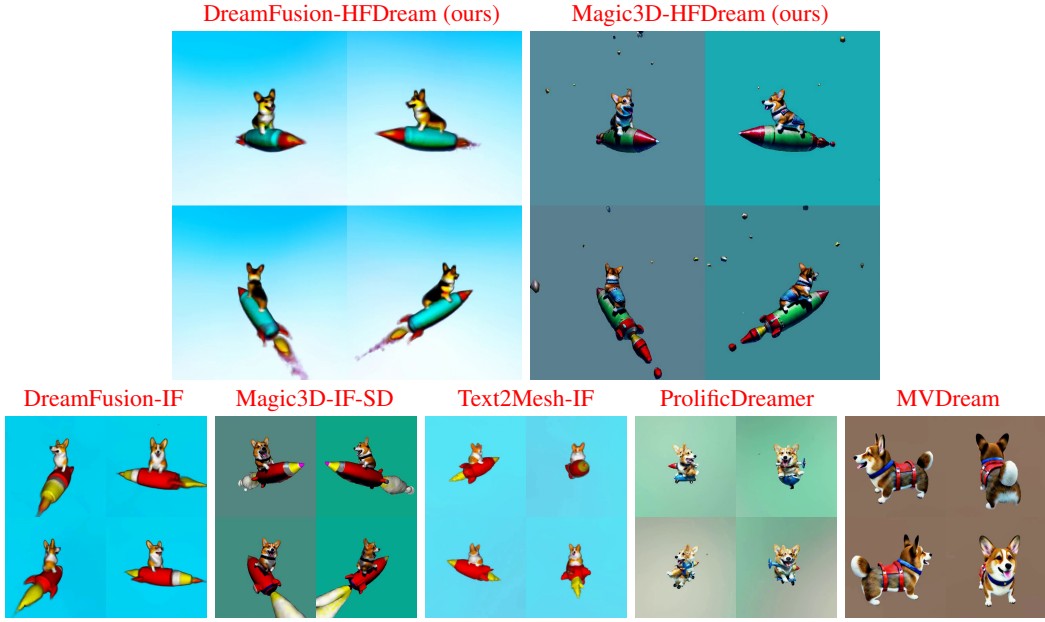

Prompt: "Corgi riding a rocket"

**Figure 13:** Comparison of HFDream to various baselines. Note that images for azimuth value 0°, 90°, 180°, 270° (front, left side, back, right side) are shown.

DreamFusion-HFDream (ours)    Magic3D-HFDream (ours)

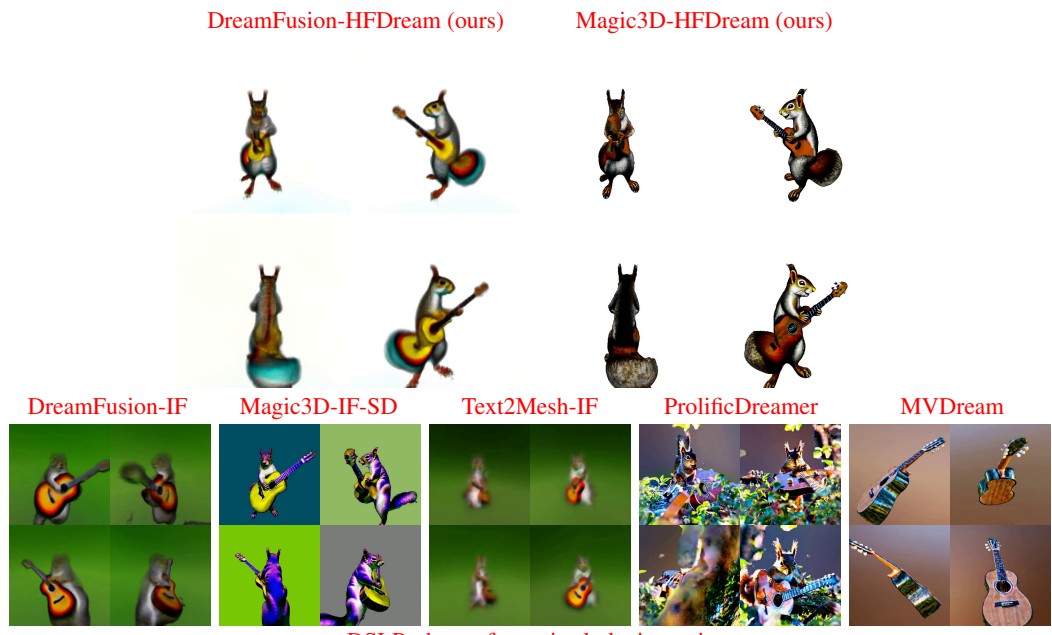

a DSLR photo of a squirrel playing guitar

**Figure 14:** Comparison of HFDream to various baselines. Note that images for azimuth value 0°, 90°, 180°, 270° (front, left side, back, right side) are shown.

DreamFusion-HFDream (ours)    Magic3D-HFDream (ours)

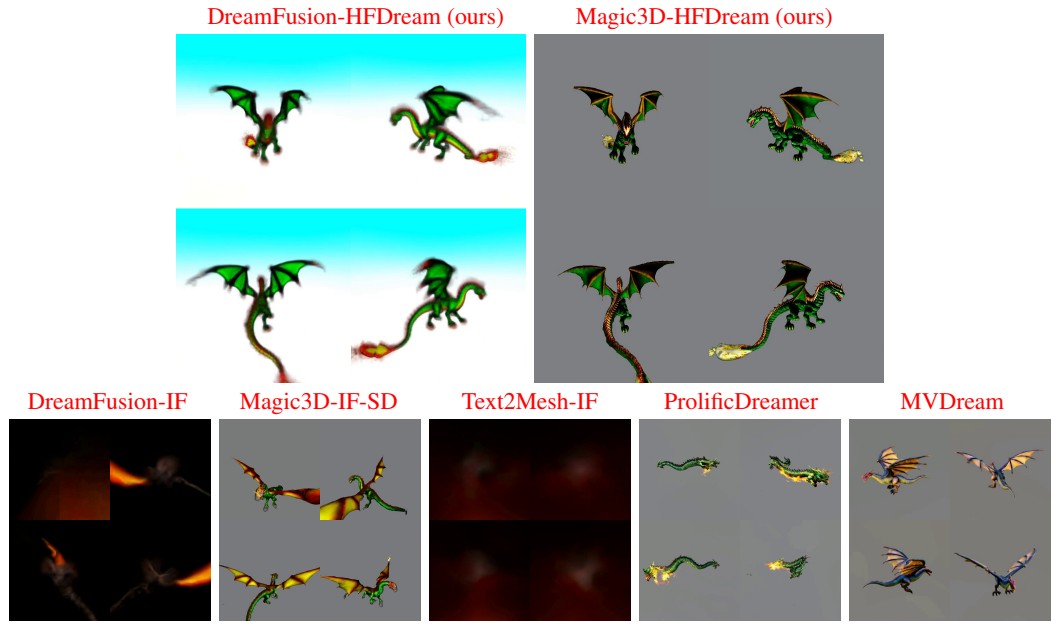

Flying Dragon, highly detailed, breathing fire

**Figure 15:** Comparison of HFDream to various baselines. Note that images for azimuth value 0°, 90°, 180°, 270° (front, left side, back, right side) are shown.

Table 5: **Text-to-3D generation comparison on random 50 prompts from DreamFusion prompts.**

|  | CLIP R-Prec. Color (↑) | CLIP R-Prec. Normal (↑) | Normalized Reward Score (↑) |
|---|---|---|---|
| DF-IF | 55.4 | 18.1 | 0.264 |
| DF-HFDream | **59.7** | **23.1** | **0.281** |

## F EVALUATION RESULTS ON DREAMFUSION EVALUATION PROMPTS

We evaluate our method on DreamFusion evaluation prompt. We measure normalized reward score, and CLIP R-Precision for RGB video rendering and normal map. As presented in Table 5, our method achieves the highest scores in all metrics.

## G EXPERIMENTS FOR HFDREAM WITH STABLE DIFFUSION

We conduct experiments with Stable Diffusion 2.1 (Rombach et al., 2022) to verify that our method can be used with other text-to-image diffusion models using human feedback. First, we update Stable Diffusion 2.1 using a reward model, which is fine-tuned with our human feedback datasets, via supervised fine-tuning (SFT) based on reward-weighted loss. As shown in Figure 16, the proposed method also improves the view-alignment of Stable Diffusion 2.1.

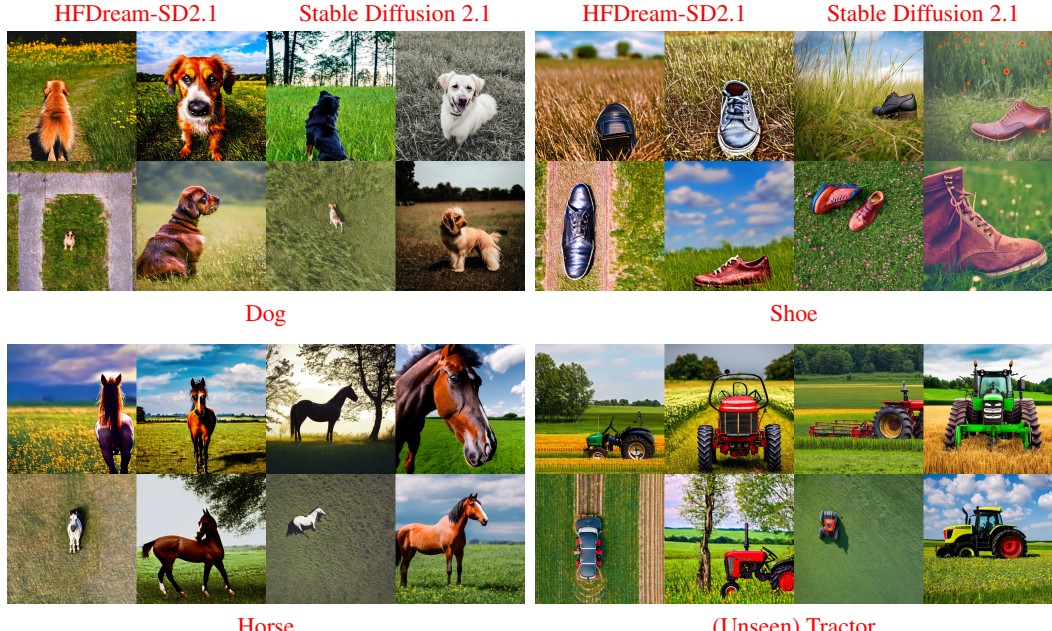

Figure 16: **Qualitative comparison** of HFDream applied to Stable Diffusion 2.1 and vanilla Stable Diffusion 2.1. View directions are: back (top left), front (top right), overhead (bottom left), side (bottom right) for each grid image.

However, supervised fine-tuning leads to a degradation in image quality (e.g., over-saturation) as also observed in several prior works (Lee et al., 2023; Fan et al., 2023). Recently, Fan et al. (2023) demonstrated that adapting reinforcement learning (RL), which involves online sample generations, can mitigate such failure cases. Inspired by this, we fine-tune Stable Diffusion 2.1 using RL methods (Black et al., 2023; Fan et al., 2023) with our reward model (denoted as HFDream-SD2.1-DDPO). As shown in Figure 17, HFDream-SD2.1-DDPO generates natural images that are also aligned with view-dependent prompts. This result shows the synergy between our method and RL fine-tuning.

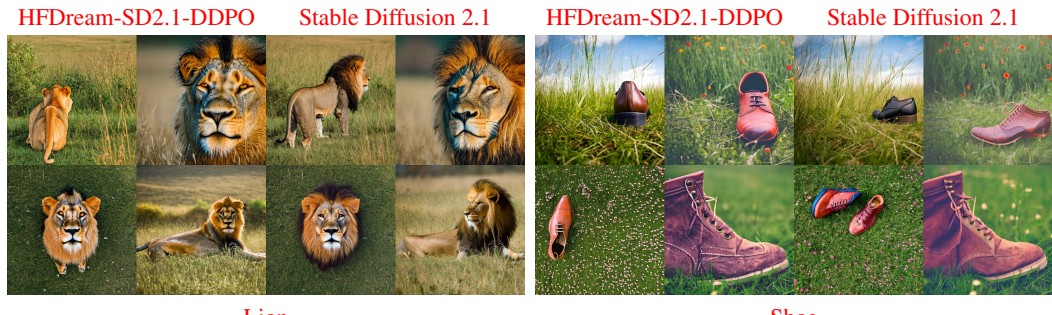

HFDream-SD2.1-DDPO    Stable Diffusion 2.1    HFDream-SD2.1-DDPO    Stable Diffusion 2.1

Lion                                          Shoe

**Figure 17: Qualitative comparison** of HFDream applied to Stable Diffusion 2.1 by using DDPO and vanilla Stable Diffusion 2.1. View directions are: back (top left), front (top right), overhead (bottom left), and side (bottom right) for each grid image.

