# OpenReview forum: "HFDream: Improving 3D Generation via Human-Assisted Multi-view Text-to-Image Models"
_ICLR.cc/2024/Conference — Submitted to ICLR 2024_

### Official Review · Reviewer_T5e1 · 2023-10-31

**Soundness:** 3 good
**Presentation:** 3 good
**Contribution:** 2 fair
**Rating:** 6
**Confidence:** 4

**Summary:**

In this paper, the authors adopt RLHF into text-to-image diffusion model and propose a new text-to-image diffusion model, termed as HFDream, which can produce viewpoint specified images with the text prompt. Furthermore, HFDream combined with DreamFusion can lead to high-quality text-to-3D generations. Specifically, they firstly collect multi-view data using human feedback, then train a reward model using these multi-view data. Subsequently, they finetune a text-to-image diffusion model with the reward model to learn multi-view image generation. Finally, DreamFusion with the finetuned diffusion model can enable high-quality 3D generation. In addition, they conduct extensive experiments to validate the proposed method.

**Strengths:**

+ The proposed method is an alternative to improve the geometric consistency of 3D generation. It adopts RLHF into diffusion models, leading to multi-view image generations and avoiding large-scale 3D assets.  Besides, it can produce pleasant 3D generations and is validated with various experiments.
+ This paper is written well and easy to follow. In addition, it provides enormous quantitative and qualitative results to validate the proposed method.

**Weaknesses:**

- To my knowledge, collecting multi-view data using human feedback is costly and somewhat difficult for some viewpoint, such as backside views. Moreover, as shown in section 4.2, it is essential to collect larger and more diverse human datasets  to improve generalization on more and unseen prompts. Thus, I raise a concern: is collecting these multi-view data cheaper than 3D assets?
- For the objects used in this paper, it is easy for human to identify their view direction, such as front view. But, there are objects for which different views is the same. Moreover, it is difficult to identify direction for 3D scenes. Thus, how to solve these problems with the proposed method?
- To validate the proposed method, the authors only use DeepFloyd-IF (pixel-based model) as their baseline text-to-image model. But, the proposed method has a generalization on diffusion models. Thus, it is better to validate the proposed method on other diffusion models, such as StableDiffusion (latent-based model).

**Questions:**

- As shown in the experiments, HFDream produces unnatural images. In detail, these images are oversaturated. In contrast, IF generates realistic images, although it does not have a good geometric consistency. Is the guidance scale high for HFDream?

---

> ### Author Response · Authors · 2023-11-18
>
> Dear Reviewer T5e1,
>
> We sincerely appreciate your review with thoughtful comments. We have carefully considered each of your questions and provide detailed responses below. Please let us know if you have any further questions or concerns.
>
> ---
>
> **[W1] Collection cost of the multi-view human feedback dataset.**
>
> We compiled a dataset with 5,000 human-labeled annotations by examining up to 10k images, a process that took about 8 hours in labeling. If we had used Amazon Mechanical Turk for this task, the cost would have been roughly \\$0.02 per annotation, totaling \\$200.
> Moreover, considering the importance of a larger and more diverse dataset, acquiring human labels for synthetically generated images is much more feasible than having experts create new, high-quality 3D assets. Collecting human labels for generated images is particularly advantageous because creating new 3D assets, which accurately reflect a given caption, is significantly more difficult and time-consuming than generating images from captions using a text-to-image model.
>
> ---
>
> **[W2] Challenges in identifying ambiguous view directions.**
>
> In practice, it is not problematic to label objects with ambiguous definitions of view directions. Specifically, we instructed the human annotators to skip the images with ambiguous view definitions and to only label the most certain ones. The detailed instructions can be found in Appendix B (Labeling Instruction).
> We also observed that the objects you referred to, which look the same from different angles, are already handled well by existing text-to-image models. Therefore, we chose to focus on objects with unambiguous view definitions, as the model struggles more with these objects to generate view-aligned images.
> We believe that human-in-the-loop learning is an effective solution for addressing view-dependent image/3D generation.
>
> ---
>
> **[W3] Lack of experiments with Stable Diffusion.**
>
> The proposed pipeline of fine-tuning text-to-image models with reward models can be used orthogonally with any text-to-image diffusion model. Still, following your suggestion, we included the results for Stable Diffusion in Appendix G. As shown in Figure 17, our method significantly improves view-alignment of Stable Diffusion 2.1, showing the generality of our approach.
>
> ---
>
> **[Q1] Over-saturation of text-to-image samples generated with HFDream**
>
> For your information, we use the guidance scale of 20 for 3D generation (same as DreamFusion-IF), and 7.5 for 2D image generation.
> We believe that the oversaturation issue is due to the supervised fine-tuning method (as observed in [1, 2]), and it can be alleviated by using sophisticated approaches such as reinforcement learning, e.g., DPOK [2] and DDPO [3].
> To support our hypothesis, we additionally conducted an experiment by applying DDPO with our proposed reward models, and found that this approach enhances view-aligned image generation without a degradation in visual quality. The experimental details and qualitative results are shown in Appendix G and Figure 18.
>
> [1] Lee et al., 2023. "Aligning Text-to-Image Models using Human Feedback." arXiv 2023.
>
> [2] Fan et al., 2023. "DPOK: Reinforcement Learning for Fine-tuning Text-to-Image Diffusion Models." NeurIPS 2023.
>
> [3] Black et al., 2023. "Training Diffusion Models with Reinforcement Learning." arXiv 2023.

---

> > ### Comment · Reviewer_T5e1 · 2023-11-22
> >
> > I have read all reviews and authors’ rebuttal. The authors resolve my concerns. And, I keep my rate.

---

> ### Author Response · Authors · 2023-11-21
> **Gentle Reminder**
>
> Dear Reviewer T5e1,
>
> Thank you very much again for your time and efforts in reviewing our paper.
>
> We kindly remind that we have only two days or so in the discussion period.
>
> We just wonder whether there is any further concern and hope to have a chance to respond before the discussion phase ends.
>
> Many thanks, \
> Authors

---

### Official Review · Reviewer_L7ZF · 2023-10-31

**Soundness:** 2 fair
**Presentation:** 2 fair
**Contribution:** 2 fair
**Rating:** 5
**Confidence:** 5

**Summary:**

This work presents a text-to-3D pipeline by using human feedback to enhance multi-view image generation. Both text-to-3D and personalized text-to-3D tasks are performed to evaluate the proposed HFDream.

**Strengths:**

++ Extensive human survey evaluation is performed.

++ It is interesting to apply HFDream for Text-to-3D DreamBooth generation task.

**Weaknesses:**

-- The whole method seems a simple human-in-the-loop engineering pipeline and is composed of several existing techniques (ImageReward and DreamFusion). The main difference is the fine-tuning of ImageReward tailed for multi-view image generation, which is somewhat not novel.

-- The experimental results are not convincing, due to the lack of many recent state-of-the-art text-to-3D baselines (e.g., Magic3D and Fantasia3D) in performance comparison.

-- I also have the concern on the human-labeled dataset which is a subset of ImageReward-v1.0 data and only has 200K pairs. So how to testify that the generalization of fine-tuned text-to-image model is not degraded?

Moreover, the evaluation prompt set is extremely small (only 61 text prompts). At least authors should report their results over the most test prompts in DreamFusion website (https://dreamfusion3d.github.io/gallery.html).

**Questions:**

The overall technical contribution is limited, and more strong baselines should be included for performance comparison.

---

> ### Author Response · Authors · 2023-11-18
>
> Dear Reviewer L7ZF,
>
> We sincerely appreciate your review with thoughtful comments. We have carefully considered each of your questions and provide detailed responses below. Please let us know if you have any further questions or concerns.
>
> ---
>
> **[W1] Limited novelty of the proposed method.**
>
> We emphasize that our key contribution is not limited to simply applying a human-in-the-loop framework to text-to-3D generation. Although the overall pipeline is intuitive and straightforward (this is rather the strength of our framework), it poses non-trivial challenges including data collection formulation and algorithmic design choices. To address them, we propose several techniques such as collecting human feedback as directions instead of preferences (Section 3.2) or using categorically normalized reward values as training weights (Section 3.3). Notably, a naive approach such as collecting human preferences similar to various prior works [1,2,3] fails to yield a high-quality reward model, leading to trivial predictions (e.g., consistently assigning low scores for rarer directions such as "back"). We believe that our work provides valuable insights to the research community.
>
> [1] Xu et al., 2023. "ImageReward: Learning and Evaluating Human Preferences for Text-to-Image Generation". NeurIPS 2023.
>
> [2] Kirstain et al., 2023. "Pick-a-Pic: An Open Dataset of User Preferences for Text-to-Image Generation." NeurIPS 2023.
>
> [3] Wu et al., 2023. "Human Preference Score v2: A Solid Benchmark for Evaluating Human Preferences of Text-to-Image Synthesis." arXiv 2023.
>
> ---
>
> **[W2] Lack of comparison to SOTA text-to-3D baselines.**
>
> We emphasize that our method is compatible with any text-to-3D techniques. We improve the text-to-image model, which serves as a backbone for text-to-3D methods, thereby enhancing the geometric consistency of the generated text-to-3D results.
> Nonetheless, in response to your suggestion, we evaluate our method in conjunction with Magic3D. Our approach, added to Magic3D (termed Magic3D-HFDream), is compared quantitatively and qualitatively against the default Magic3D-IF-SD setup. As presented in Table 2, we measure CLIP-R precision score and normalized reward score values, finding significant improvement in both metrics.
> Moreover, we conducted a comparative analysis against several SOTA baselines, including MVDream, ProlificDreamer, Magic3D, TextMesh, and DreamFusion-IF. This comparison was done using demos from the MVDream project website. For instance, in the 'Corgi riding a rocket' example in Figure 13, we observed that while baseline methods compromise either prompt alignment (MVDream, in which the rocket is missing) for geometric consistency or vice versa (DreamFusion-IF, Magic3D-IF-SD, TextMesh, ProlificDreamer), our Magic3D-HFDream demonstrates consistently better results in both aspects. The visual comparison can be found in Appendix E.3, and the rendered video is available in the updated supplementary materials and in this [Google Drive link](https://drive.google.com/drive/u/3/folders/13upykprOflx3K6gLpNdTpVIgbwhpSt1d).
>
> ---
>
> **[W3] Concern about the human-labeled dataset and the generalization of the fine-tuned text-to-image model.**
>
> We would like to clarify your potential misunderstanding. Our dataset is distinct from ImageReward-v1.0 and is not a subset of it. We created our dataset by generating images with text-to-image models and then collecting human feedback on valid view direction,  as described in Section 3.1. We labeled about 5k images in total.
>
> Regarding generalization, we remark that both the reward function and the fine-tuned text-to-image model are evaluated on unseen text prompts (describing unseen objects). As shown in Figure 6, the reward model shows great reliability for all directions, and also generalizes very well, with minimal accuracy drop even for unseen objects. Table 1 also shows the clear improvements in view-dependent image generation for unseen objects.
> Additionally, we use LoRA [1] to help retain the generalization of the fine-tuned text-to-image model.
>
> [1] Hu et al., 2021. "LoRA: Low-Rank Adaptation of Large Language Models." ICLR 2022.
>
> ---
>
> **[W4] Small evaluation prompt set size.**
>
> Our main goal is to evaluate the generalization of the text-to-image model systematically. Specifically, the reason we evaluate two different sets of prompts is to verify HFDream generalizes well to prompts with unseen objects.
> Additionally, we believe that running 244 generations total with 61 different prompts is enough to test the effectiveness of our method, given our goal was to demonstrate that our method can consistently perform reliable generation with each of the random seeds.
> Nevertheless, following your suggestion, we evaluate HFDream on DreamFusion prompts as well, and present the results in Appendix F. We chose 50 different prompts among the 411 prompts, due to the tight time schedule, on top of additional prompts that were used in MVDream.

---

> ### Author Response · Authors · 2023-11-21
> **Gentle Reminder**
>
> Dear Reviewer L7ZF,
>
> Thank you very much again for your time and efforts in reviewing our paper.
>
> We kindly remind that we have only two days or so in the discussion period.
>
> We just wonder whether there is any further concern and hope to have a chance to respond before the discussion phase ends.
>
> Many thanks, \
> Authors

---

> > ### Comment · Area_Chair_kpX9 · 2023-11-22
> >
> > Hi Reviewer L7ZF,
> >
> > Can you read the rebuttal and check if it resolves your concerns? Thanks.
> >
> > Your AC

---

### Official Review · Reviewer_PzSD · 2023-11-02

**Soundness:** 3 good
**Presentation:** 3 good
**Contribution:** 2 fair
**Rating:** 5
**Confidence:** 5

**Summary:**

This paper proposes to enhance text-to3D generation by learning from human feedback for generating desired views. It collects a human-labeled dataset to train a reward model and adopts this model to fine-tune a diffusion model, and this reward model can help the diffusion model generate the image aligned with the viewpoint text.

**Strengths:**

1. The whole idea is Intuitive and effective.
2. The reward model can help the model generate images that match the viewpoint text.
3. The paper is well written.

**Weaknesses:**

My main concern is the motivation of this work. Indeed, Dreamfusion has a multi-face problem since the diffusion can not generate the image aligned with the camera. There are two main methods that can solve this problem. First type: we can train a camera-aware diffusion model like Zero123 and MVdreamer[1]. These models can generate an image aligned with the camera.  These methods use the camera parameter as guidance, which is much more correct than viewpoint text since the viewpoint text only has three settings (front, side, and over). Second type: We can introduce the Controlnet [2] or the mesh as the 3D prior [3] to solve the multi-face problem.
So, what is the advantage of human feedback? The whole process is time-consuming and cumbersome, and I found the results were not very well either.  Meanwhile, in addition to the multi-face problem, the proposed method still has the same issues with Dreamfusion: Over smooth and over-saturated.

[1] Shi, Yichun, et al. "Mvdream: Multi-view diffusion for 3d generation." arXiv preprint arXiv:2308.16512 (2023).
[2] Han, Xiao, et al. "HeadSculpt: Crafting 3D Head Avatars with Text." arXiv preprint arXiv:2306.03038 (2023).
[3] Chen, Rui, et al. "Fantasia3d: Disentangling geometry and appearance for high-quality text-to-3d content creation." arXiv preprint arXiv:2303.13873 (2023).

**Questions:**

See the weakness.

---

> ### Author Response · Authors · 2023-11-18
>
> Dear Reviewer PzSD,
>
> We sincerely appreciate your review with thoughtful comments. We have carefully considered each of your questions and provide detailed responses below. Please let us know if you have any further questions or concerns.
>
> ---
>
> **[W1] Concerns about motivation.**
>
> It is important to note that our framework (i.e., learning from human feedback) is time- and resource-efficient approach for collecting high-quality data compared to alternative approaches; (a) Training a camera-pose aware diffusion model requires 3D asset data which is known to sacrifice prompt fidelity at the cost of geometric consistency, due to distribution shift. (b) Adopting a ControlNet-like approach relies on additional priors and is limited to face landmarks only [1]. Finally, we remark that our method is orthogonal to other approaches and can be used together for even better performance.
>
> [1] Han et al., 2023. "HeadSculpt: Crafting 3D Head Avatars with Text." arXiv 2023.
>
> ---
>
> **[W2] Unsatisfactory quality of the text-to-3D generation results.**
>
> We emphasize that our method is compatible with any text-to-3D techniques since we fine-tune the text-to-image model, which serves as a backbone for text-to-3D methods. Nevertheless, to address quality concerns, we applied our method to the more advanced backbone, Magic3D (termed Magic3D-HFDream), and compared against the default Magic3D (termed Magic3D-IF-SD) by following the suggestions of Reviewer L7ZF. Additional qualitative and quantitative results using Magic3D-HFDream are available in Appendix D of the revised draft. As shown in Table 4, our method consistently improves the CLIP-R precision score and the normalized reward score, and alleviates the Janus problem as shown in Figure 8. We also provide qualitative results using prompts and demos from the [MVDream demo website](https://mv-dream.github.io/test_2.html). As shown in Appendix E.3, Figure 13, Figure 14, and Figure 15, the baseline methods either sacrifice prompt alignment (MVDream, in which the rocket is missing) or geometric consistency (DreamFusion-IF, Magic3D-IF, TextMesh, ProlificDreamer), while our result (Magic3D-HFDream) achieves better results in both aspects. The rendered video file is attached in the updated supplementary material and can also be found in the [Google Drive link](https://drive.google.com/drive/u/3/folders/13upykprOflx3K6gLpNdTpVIgbwhpSt1d).
>
> ---
>
> **[W3] Over-saturation and over-smoothness of the text-to-3D generation results.**
>
> It is important to note that our main focus is on addressing the multi-face problem, which is the most critical challenge in current text-to-3D generation methods, as highlighted by many prior works [1, 2, 3]. While we agree that over-saturation and over-smoothness are also important issues, we think that these issues are not of our major focus and can be mitigated by combining our method with more advanced methods, such as Magic3D. For example, when combined with Magic3D, our method generates 3D outputs with reduced over-saturation and over-smoothness (see Figure 8, Appendix E.3 and [Google Drive link](https://drive.google.com/drive/u/3/folders/13upykprOflx3K6gLpNdTpVIgbwhpSt1d)).
>
> [1] Poole et al., 2023. DreamFusion: Text-to-3D using 2D Diffusion.
>
> [2] Shi et al., 2023. MVDream: Multi-view Diffusion for 3D Generation.
>
> [3] Wang et al., 2023. ProlificDreamer: High-Fidelity and Diverse Text-to-3D Generation with Variational Score Distillation.

---

> ### Author Response · Authors · 2023-11-21
> **Gentle Reminder**
>
> Dear Reviewer PzSD,
>
> Thank you very much again for your time and efforts in reviewing our paper.
>
> We kindly remind that we have only two days or so in the discussion period.
>
> We just wonder whether there is any further concern and hope to have a chance to respond before the discussion phase ends.
>
> Many thanks, \
> Authors

---

> > ### Comment · Area_Chair_kpX9 · 2023-11-22
> >
> > Hi Reviewer PzSD,
> >
> > Can you read the rebuttal and check if it resolves your concerns? Thanks.
> >
> > Your AC

---

### Author Response · Authors · 2023-11-18

Dear reviewers and AC,

We sincerely appreciate your time and effort you have dedicated to reviewing our paper. Your insights have been invaluable.

As reviewers highlighted, we believe our paper provides an effective (PzSD, T5e1) method to improve the view-aligned generation of text-to-image models with human feedback, validated with extensive evaluations (L7ZF, T5e1) followed by a clear presentation (PzSD, T5e1).

In addressing your feedback, we have carefully revised and enhanced our manuscript with the following additional sections:

* Comparison to state-of-the-art baselines and analysis (in Appendix D, Appendix E.3).
* Evaluation results for evaluation prompts of DreamFusion (in Appendix F).
* Experiments with Stable Diffusion and alternative approaches for fine-tuning (in Appendix G)

For ease of review, these updates are temporarily highlighted in red.

We believe that HFDream can be a useful addition to the ICLR community, in particular, due to the above revision helping us better deliver the effectiveness of our method.

Thank you very much!

Authors.

---

### Meta-Review · Area_Chair_kpX9 · 2023-12-06

**Metareview:**

The paper receives mixed borderline reviews. Reviewers appreciate the intuitive design and well-written paper. However, reviewers also raise concerns about the motivation and the novelty. In particular, the paper is mostly a simple adoption of RLHF to resolve the Janus problem commonly seen in text-to-3D models. While there are some non-trivial design choices (such as human feedback forms and categorically normalized reward), they do not seem significant enough to get published at the ICLR venue. The authors are encouraged to provide more insights and design corresponding experiments which could be potentially more interesting to the community.

**Justification For Why Not Higher Score:**

The paper is overall a simple adoption of RLHF, which no major modifications or particular insights.

**Justification For Why Not Lower Score:**

N/A

---

### Decision · Program_Chairs · 2024-01-16

Reject